# Porous Structural Microfluidic Device for Biomedical Diagnosis: A Review

**DOI:** 10.3390/mi14030547

**Published:** 2023-02-26

**Authors:** Luyao Chen, Xin Guo, Xidi Sun, Shuming Zhang, Jing Wu, Huiwen Yu, Tongju Zhang, Wen Cheng, Yi Shi, Lijia Pan

**Affiliations:** Collaborative Innovation Center of Advanced Microstructures, School of Electronic Science and Engineering, Nanjing University, Nanjing 210093, China

**Keywords:** microfluidic, porous structure, biomedical, biosensor

## Abstract

Microfluidics has recently received more and more attention in applications such as biomedical, chemical and medicine. With the development of microelectronics technology as well as material science in recent years, microfluidic devices have made great progress. Porous structures as a discontinuous medium in which the special flow phenomena of fluids lead to their potential and special applications in microfluidics offer a unique way to develop completely new microfluidic chips. In this article, we firstly introduce the fabrication methods for porous structures of different materials. Then, the physical effects of microfluid flow in porous media and their related physical models are discussed. Finally, the state-of-the-art porous microfluidic chips and their applications in biomedicine are summarized, and we present the current problems and future directions in this field.

## 1. Introduction

Microfluidics, also known as lab-on-a chip (LOC), is a device that integrates biomedical analysis experiments, including sample preparation, separation, reaction, detection, etc. [1,2,3]. Microfluidic chips can achieve rapid results with a small amount of samples through the precise control of fluids. Since scientists developed micron-scale gas chromatography columns on silicon chips in 1979, microfluidic chips have undergone considerable development and have been widely used in the fields of biology, chemistry and medicine [4,5,6,7,8,9,10,11]. At present, microfluidic chips are still facing challenges such as large interference and difficult separation. Therefore, various innovative technologies based on preparation methods and structure have been proposed to achieve better performance. For example, the combination of intelligent communication devices and microfluidic chips has been proposed to enable easier real-time healthcare detection [12,13,14,15]. The combination of spinning technology and microfluidic chips is applied to flexible wearable devices [16,17,18,19,20].

In addition to device preparation methods and structural innovations, changing the structure of materials can also play a role in improving the sensing performance, stability and sensitivity of microfluidic devices. Porous materials, with their high specific surface area, good permeability, low relative density, and high specific strength, are widely used in microfluidic chips for applications such as cell separation, cell culture, and real-time biomedical detection [2,21,22,23,24]. Due to its high specific surface area, the porous structure provides a basis for achieving a higher capture rate and provides another idea for improving detection sensitivity. The original and most widely used porous structure material is paper. As a material with a natural porous structure, paper has the advantages of low cost and light weight. In 1952, Martin and Synge invented paper-based chromatographic separation technology and won the Nobel Prize in Chemistry [25]. Since then, paper-based microfluidic devices have been widely used in detection and medical treatment, and there are many related literatures reviewing their related applications [1,8,26]. In addition, with the development of microelectronics technology in recent years, another porous material commonly used in the field of biomedical detection is polymers such as PDMS. Compared with traditional glass and silicon materials, PDMS has the advantages of elasticity, castability, and surface chemical properties that can be modified, and it is increasingly used in microfluidic devices [27,28,29,30]. Multi-well PDMS screening plates are commonly used in cell sorting and cell culture. There are also novel flexible porous materials such as hydrogels and textile fabrics for microfluidic applications, which we will describe in detail later in the article.

In this review, various microfluidic devices based on porous structure will be discussed by material, including their characteristics and fabrication methods. In order to better understand the flow mechanism of fluids in porous structured microfluidic channels, in the next section, relevant physical principles and the fluid dynamics related models of porous structures are discussed. Then, the application of microfluidic devices based on porous structures in biomedical diagnostics will be discussed in scenarios (Figure 1). Finally, we will summarize and look forward to the future development of porous microfluidic devices.

## 2. Different Materials and Preparation Methods for Microfluidic Devices Based on Porous Structures

The most outstanding feature of the microfluidic system is its ability to analyze and manipulate microliters of fluid in channels with diameters of 10–100 μm [41,42]. A porous structured microfluidic system with small thermal mass, high mass transfer efficiency, high specific surface area, good permeability, low relative density and high specific strength can effectively improve the flux rate and sensitivity of the device [43,44,45,46,47,48,49]. The diversity of porous materials stems from a wide range of preparation methods [50,51,52,53,54]. Because of their controllable pore size and porosity, porous materials offer a wide range of applications for microfluidic systems [9,55,56,57,58,59].

Initially, with the development of the silicon-based industry, the first generation of microfluidic chips was created and prepared on silicon/silicon dioxide substrates by traditional microfabrication methods [60,61]. The typical microfabrication process consists of three steps: lithography and development, etching to form microchannels, and bonding to assemble the microfluidic chip. Silicon-based microfluidic chips were first developed and have advantages such as high thermal conductivity, special optical properties and electro-osmotic stability. However, their high cost, complex process flow, time consuming nature, rigidity and gas impermeability and other shortcomings have become obstacles to the large-scale application of health monitoring and wearable biomedicine [10,60,62,63,64].

To meet the market demand for the mass production of microfluidics for biomedical and clinical applications, chips are required to be flexible, breathable, low cost, disposable, and environmentally friendly. Therefore, microfluidic devices based on porous three-dimensional structures of paper, soft elastic materials and textile fabrics have been created [1,37,65,66].

### 2.1. Fabrication of PDMS and PMMA-Based Microfluidics

Synthetic polymers are ubiquitous materials used in a wide variety of applications because of their structural and mechanical properties [67]. The application of polymers as a discontinuous dielectric structure in wearable devices offers a wide range of applications in health monitoring and medical fields [27,68,69,70]. The rapid development of porous polymers in the last few years has been attributed to the continuous development of modern organic synthesis, advanced polymerization techniques and nanotechnology [71].

Polydimethylsiloxane (PDMS) has been widely used in many research fields because of its excellent properties such as easy fabrication, high flexibility, and thermal stability [72]. PDMS is a commonly used material in microfluidics. The layer-by-layer process allows the fabrication of 3D microfluidic molds for casting PDMS. Two-dimensional (2D) PDMS layers can be combined together to form 3D structures. Techniques such as the sacrificial template method and soft lithography are also commonly applied to the preparation of PDMS materials with porous structures [29,73,74]. UV modifications are also frequently used on PDMS to modify the surface properties by tuning their intensity [75,76]. These modifications alter the wettability by adding functional groups, creating thin layers on the PDMS surface and significantly enhancing the potential for microfluidic applications.

Due to the simplicity, safety and low cost of preparation, the sacrificial template method is commonly used to obtain porous PDMS sponges [36,77]. The successful fabrication of porous PDMS using sugar as a template and water as a solvent was first reported by Choi et al. [35]. PDMS sponges composed of porous, interconnected three-dimensional frameworks were prepared by using cubic sugars of different particle sizes as templates (Figure 2a). This structure of PDMS sponge can largely bend and recover the original shape almost perfectly, thus greatly improving the stability of flexible microfluidic devices. Similarly, Zhao et al. prepared porous PDMS sponges using NaCl instead of cubose as a template [78]. Yu et al. used citric acid monohydrate (CAM) as a template and ethanol as a solvent to make 3D interconnected porous PDMS sponges with high porosity, flexibility and superwettability [72]. Other early conventional methods for preparing porous microfluidic chips include photolithography and etching techniques [79,80,81,82]. However, the dies fabricated by complex processes such as photolithography are limited by flexibility, and the die pattern cannot be changed once it is fabricated. At the same time, the traditional photolithography and etching methods have the disadvantages of high cost, complicated process and being time consuming. In contrast, laser micromachining and printing technology becomes a better choice because of its simplicity, flexibility, controlled aperture size, speed and direct writing of different collection shapes [83,84,85]. Chen et al. report on printing protocols for PDMS with porous structures printed using liquid dispensers, flow control capabilities, flow delay mechanisms, and applications in sequential delivery [30]. Montazerian et al. prepared a PDMS porous scaffold with triply periodic minimal surfaces structure with radial gradient porosity by combining 3D printing technology [36]. This porous structure of PDMS material is highly elastic, water permeable and biocompatible (Figure 2b).

Another material commonly used to prepare microfluidic devices is PMMA, which is a thermoplastic material that has better mechanical properties than PDMS and can maintain good initial morphology under mechanical stress conditions. In addition, PMMA has the advantages of stable chemical properties, optical transparency and low cost. The common methods used to process PMMA are hot embossing, injection molding and direct laser writing. Bouchard et al. fabricated periodic patterns with linear and dotted geometrical features on stainless steel surfaces with sizes ranging from 1.7 to 900 µm by combining different laser-based processes, namely direct laser engraving (DLE), direct laser writing (DLW), and direct laser interference patterning (DLIP). Then, the fabricated layered geometries are transferred to the PMMA surface by plate-to-plate thermal embossing [86]. Volpe et al. developed and tested a new intelligent procedure for the rapid fabrication of PMMA LOC prototypes for cell capture by using femtosecond laser technology for microchannel mechanical micro-milling of the inlet and outlet connections and thermal bonding to complete the device [87]. The snake microchannels are then directly biofunctionalized by fixing capture probes, which can distinguish between cancer and non-cancer cells without labeling. The device is useful for the label-free capture and identification of tumor cells from blood cells.

As the traditional materials for fabrication microfluidic devices, PDMS and PMMA play an important role in microfluidics due to their simple processing (compared to silicon and glass), excellent mechanical properties and low preparation cost, and they will have a broader commercial product and market demand in the future.

### 2.2. Fabrication of Paper-Based Microfluidics

Paper has unique advantages over other materials in terms of low cost, flexibility and self-driven fluid pumping, thus making it widely used in various fields of wearable devices and human health monitoring [88,89,90,91,92]. Paper-based microfluidics is a fast-growing field because the inherently porous structure of paper provides a separate site for transporting liquids by capillary forces [1,93,94,95]. In recent years, with the rapid development in the field of materials, the definition between traditional paper and flexible films has become blurred, and some flexible films with flexible or porous structures have been defined as paper [96]. Some treatment and processing of the microstructure of the paper is usually necessary to create paper-based microfluidic devices, including wax printing, printing, photolithography, etching, chemical alteration of the fiber surface, and other methods [97,98,99,100].

Photolithography was used to create the first paper-based microfluidic devices and is still the most common preparation method due to its advantages such as high resolution and accuracy [3]. Photolithography begins by covering the paper with photoresist and forming a proper patterned cross-linked photoresist by using a photomask. Finally, the remaining photoresist is eliminated to form patterned microstructures. Dong et al. proposed a microfluidic paper-based analytical device fabricated by UV curing using aqueous polyurethane acrylate (PUA) to pattern the filter paper [101] (Figure 3a). Wax is a non-toxic, disposable, low-cost and easily patterned hydrophobic substance. Therefore wax printing has become a popular process for preparing disposable paper-based microfluidic devices [100,102,103]. A sheet of paper with a wax print is heated to allow the wax to flow and allow it to reach the thickness of the paper. This creates a completely impermeable hydrophobic barrier with a hydrophilic zone inside it shaped like a wax print pattern. Carrilhoo et al. elaborate on the wax-printing technique, where a roll of paper is first passed through a wax printer, then heated through an oven, and finally through an inkjet printer, where reagents for testing or other applications are printed in the test area [104]. Mani et al. prepared low-cost paper-based microfluidic devices for patterning using a wax-printing method. The microfluidic structures were made by simply patterning them. Equipping the device with RuPVP/DNA/enzyme spots can be used to demonstrate screening for genotoxic compounds in water, food and smoke [105]. The integration of printed electronics and microfluidics on paper is still in the early stages of development. Hamedi et al. added ink to the microchannels in combination with electronic printing techniques to form internal paper conductors. The ink covered the cellulose fibers without clogging the pores of the paper and kept the channels hydrophilic [37] (Figure 3b). Although devices for paper-based microfluidic devices have experienced rapid development in recent years, there are still many key challenges to overcome such as resolution, biocompatibility and precision.

### 2.3. Fabrication of Three-Dimensional Hydrogels and Textile Fabrics-Based Microfluidics

Although significant progress has been made in the past few years in the fabrication of PDMS flexibles, 2D papers and membranes, they still suffer from the drawbacks of limited biocompatibility and poor mechanical properties, which still hinder the wide application of microfluidic devices in biomedical applications. Three-dimensional cross-linked networks combined with conductive hydrogels and three-dimensional electrotextiles have received widespread attention in the field of wearable microfluidic devices using their unique tunable mechanical flexibility, high biocompatibility and excellent electronic properties [106,107]. The current preparation of conductive hydrogels is generally based on co-networking, self-assembly and blend techniques [108,109,110,111]. Joo et al. prepared conductive hydrogels composed of poly(3,4-ethylenedioxythiophene), in which polystyrene sulfonate (PEDOT:PSS) acted as the main conductive pathway for electrical signal transmission and the hydrogel cross-linked polymer network was highly stretchable [38] (Figure 4a). Some ions are added to the hydrogel, which increases the electrical conductivity of the conductive hydrogel by increasing ion migration due to the presence of large amounts of water in the hydrogel network (Figure 4b). Zhi et al. developed cross-linked polyacrylamide hydrogels with electrical conductivity provided by vinyl hybridized silica nanoparticles [33]. Crosby et al. proposed a simple one-step embossed lithographic patterning method to form flexible and highly stretchable patterned composite fabrics by an embossed photolithography procedure [39] (Figure 4c). By regulating the arrangement of the fibers and the three-dimensional shape of the fabric, the anisotropy of the material will be regulated (Figure 4d). This property of the fabric maintains a higher degree of directional bending flexibility than conventional flexible materials. Chen and colleagues synthesized porous graphene fibers by microfluidic orientation strategy [32]. The fibers have a uniformly dense porous network, excellent flexibility and superior electrical conductivity, providing excellent applications for wearable medical electronics.

Table 1 summarizes the above-mentioned methods for the preparation of porous structures of different materials and provides some additional information. For each method, we have listed the advantages and limitations, and readers can choose the appropriate method according to their need in different scenarios.

## 3. Relevant Principles in Microchannels and Hydrodynamically Relevant Models for Porous Structures

Understanding the flow behavior of microfluids in porous structures has important implications not only for theory but also for practical applications. Usually, the dynamics of microfluids obeys the linear Stokes equations, but due to the existence of porous structures, the variable interfacial tension and specific boundary conditions change to produce nonlinear behavior [60,136,137,138]. Fluid dynamics can only provide some guidance at this point, while the reality of the problem becomes more complex. Therefore, some complex real-world problems can be solved by some flexible application platforms and simulations.

### 3.1. Inertial Effect of Microfluidics

From the above, it can be seen that the flow of microfluidic devices can be considered as Stokes flow when the Reynolds number of the flow field is neglected [139]. When inertia and viscosity effects are taken into account, the particle flow will no longer conform to the Stokes equations, thus becoming more complicated [140]. The inertial effect was first discovered in macroscopic pipes, where initially millimeter-sized suspended particles randomly distributed in a circular pipe (about 1 cm) migrated laterally to focus on a ring with a radius 0.6 times the radius from the center of the pipe [141]. Under moderate Reynolds number conditions, the migrating ions in the circular channels form the Segre–Silberberg annulus due to the symmetry of the rings [142]. In a curved channel, where the fluid in the central region has a higher velocity than in the wall region, the Poiseuille flow has a velocity curve similar to a parabola in the main flow direction. When the particles pass through the curved closed channel, in order to satisfy the mass conservation, the fluid near the outer wall is recirculated inward due to the centrifugal pressure gradient, and two vortices with opposite rotational directions are generated in the cross-sectional direction (lab on a chip) (Figure 4). This fluid motion profile for cross-sectional quadratic flow was first proposed by Dean and is characterized by the dimensionless coefficient De:(1)$$D_{e}=R_{e}\sqrt{\frac{H}{2R}}$$
where R is the curvature radius of the channel, and H is the hydraulic diameter [143,144]. Dinler et al. simulated the fluid motion behavior of particles under the action of the Dean vortex and predicted the focusing bandwidth for particles of different sizes [145]. Due to the presence of secondary flow, the particles in the curved channel are also affected by the Dean drag force: FD∝ρU2apH2R−1, where U is the velocity of the fluid, H is the hydraulic diameter, ap is the diameter of the particle, ρ is the density of the fluid, and R is the curvature radius of the channel. Gossett believes that at low velocities, the particles are subject to the interaction of transverse inertial lift and the Dean drag force [146].

### 3.2. Electrorheological Effect of Microfluidics

When the fluid and the particles dispersed in the fluid are non-conductive or slightly conductive, an applied electric field is applied to the dispersion and the particles will be electrodepolarized due to the different dielectric constants of the objects. The induced dipole moment can be expressed as:(2)$$\vec{P}=\frac{\varepsilon_{s}-\varepsilon_{\iota }}{\varepsilon_{S}+2\varepsilon_{\iota }}R^{3}\vec{E_{l}}={\beta R}^{3}\vec{E_{l}}$$
where εs and εl denote the complex permittivity of solid particles and liquids, respectively, and R denotes the radius of the sphere, β is the Claussius–Mossotti (CM) factor, and El is the field at the local field. The particles will tend to aggregate and form chains along the direction of the applied field, which is a phenomenon that is the increasing viscosity behavior exhibited by colloids when sheared in the direction perpendicular to the electric field [147]. Earliest, the model of the induced dipole–dipole interaction proposes that for the same system of dielectric microspheres dispersed in an insulating fluid, the lowest energy state is where the microspheres aggregate and form columns along the applied field direction, and the microstructure inside the columns can be predicted from the dipole–dipole interactions [148]. In recent years, Yethiraj et al. discovered the phase behavior of colloidal systems with tunable interparticle interactions under an external electric field by confocal microscopy [149]. Hynninen et al. used simulations to calculate the Helmholtz free energy of a similar system [150], and the simulation results obtained were in good agreement with the experimental results of Yethiraj et al. Although many problems of dielectric electrorheological fluids can be obtained and solved from simulations and experiments, they cannot provide an overview of dielectric electrorheological mechanisms. The microstructural changes in the rheological properties induced by the electric field should also be accompanied by electrical manifestations. In the presence of an electric field, the effective dielectric constant of the system reflects anisotropy:(3)$$\langle \vec{D}\rangle ={\bar{\varepsilon }}_{\mathrm{eff}}\langle \vec{E}\rangle$$
(4)$${\bar{\varepsilon }}_{\mathrm{eff}}=\left(\begin{array}{ccc} {\bar{\varepsilon }}_{\mathrm{xx}} & {\bar{\varepsilon }}_{\mathrm{xy}} & {\bar{\varepsilon }}_{\mathrm{xz}}\\ {\bar{\varepsilon }}_{\mathrm{yx}} & {\bar{\varepsilon }}_{\mathrm{yy}} & {\bar{\varepsilon }}_{\mathrm{yz}}\\ {\bar{\varepsilon }}_{\mathrm{zx}} & {\bar{\varepsilon }}_{\mathrm{zy}} & {\bar{\varepsilon }}_{\mathrm{zz}} \end{array}\right)$$

The concept of an effective dielectric constant is based on the nature of the interaction of electromagnetic waves with non-uniform materials. When the wavelength is large, the microstructure cannot be resolved, and the composite material appears homogeneous to the probe wave, whose electromagnetic response is completely captured by the effective permittivity tensor [151]. The Gibbs free energy density of the fluid–solid complex with the participation of the dielectric constant tensor is:(5)f=−18πE→⋅Reε¯eff⋅E→−TS=−18πReε¯zzE2−TS
where S is the entropy and Reε¯zz denotes the real part in parentheses. The method of variable effective permittivity allows us to obtain the dielectric electrorheological ground state structure, thus providing a basis for quantitative assessment of the rheological properties that may arise from structural deformation [152,153].

### 3.3. Fluid Behavior in Porous Media

Paper has the advantages of being inexpensive, being easy to obtain and having a high specific surface area, making it the most common medium in porous [26,37]. Paper consists essentially of a random distribution of cellulose, and the flow of liquid in paper is dominated by capillary [1,154]. The Lucas–Washburn (L-W) model considers the porous system as a bundle of parallel rigid capillaries with an average cross-section. Its equation is expressed as:(6)$$x\left(t\right)=\sqrt{\frac{r\sigma t\;\cos\theta }{2\mu }}$$
where σ is the surface tension (gas–liquid), μ is the dynamic viscosity of the liquid, r is the capillary radius, and θ is the contact angle between the capillary wall and the liquid [155]. When used to describe the fully saturated wetting flow of a liquid through a paper system, it can only be used in the case of one-dimensional flow of a single homogeneous porous system. Darcy’s model was used to capture capillary-driven transport through porous structures in a multidimensional flow case [156]. The volumetric flow rate through a porous paper medium under this model is:(7)$$Q=-\frac{\mathrm{kA}}{\mu x}\Delta P$$
where k is the permeability, μ is the dynamic viscosity of the fluid, A is the cross-sectional area, and ΔP is the pressure difference across the length x [157]. However, both models are based on the assumption of full saturation by the wetting front of the liquid. Buser et al. proposed a model for flow in partially saturated porous paper media by means of the Richards equation:(8)$$\frac{\delta \theta }{\delta t}=\frac{\delta }{\delta z}\left[K\left(\theta \right)\frac{\delta H\left(\theta \right)}{\delta z}\right]$$
where K is the hydraulic conductivity and H is the pressure head; the equation relates the change in saturation of the porous medium to the pressure head gradient [158]. This model can be thought to accurately describe the fluid flow in porous media.

Early theoretical analysis of flow in porous media used a coarse-grained approach, which employs macroscopic properties. For Newtonian flow, Darcy et al. proposed a relationship between the pressure drop per unit length ΔP/L and the average velocity V:(9)$$\frac{\Delta P}{L}=\frac{\eta V}{K}$$
where η is the viscosity of the fluid and the permeability K is a constant that depends on the properties of the medium [159]. However, this equation is limited to describe simple flows and Newtonian flows on a macroscopic scale. For the flow of a fluid through a porous structure, the classical problem of flow around a cylinder can be a benchmark for non-Newtonian fluid dynamics. When studying the flow of fluids in different geometries, two-period cylindrical arrays are usually used. For example, two-dimensional flow in two-period cylindrical arrays was studied by Alcocer and Singh et al. They demonstrated a non-monotonic dependence of permeability on aspect ratio [160,161].

In almost all porous media, the main driving force of the fluid is the capillarity. The lateral flow of a fluid in a microfluidic channel usually follows the Washburn equation:(10)$$l=\sqrt{\frac{\gamma \mathrm{COS}\theta }{2\eta }}\mathrm{rt}$$
where γ is the surface tension, θ is the contact angle, r is the average pore radius, t is the time, η is viscosity of the fluid, and l is the distance. The equation shows that when the temperature and channel width are equal, the distance traveled by the fluid is proportional to the square root of time [155].

### 3.4. Other Relative Models

Based on the previous fluid dynamics, the researchers developed further physical models in different application scenarios to make more accurate physical simulations and calculations of the flow properties of liquids in porous structures.

Chang et al. proposed that the limited accuracy of the Washburn equation is mainly due to the internal cavities of the cellulose fibers that compose the paper [162]. They conducted a combined experimental and theoretical study, where experimental measurements of the internal structure of cellulose fibers were carried out and a mathematical model of liquid absorption was developed by considering the flow through the inner pores of the fibers.

In this model, the relationship between the length of the liquid absorbed by the cylindrical tube *ll*) and the time of inflow into the slit (*t*) is:(11)$$l^{*}\left(t^{*}\right)l^{*\prime }\left(t^{*}\right)+\frac{1}{2}\psi f\left(t^{*}\right)=\frac{1}{2}$$
where
(12)$$l^{*}=l/l_{c}$$
(13)$$t^{*}=t/t_{c}$$


(14)
$$f\left(t^{*}\right)=\left\{\begin{array}{c} \int_{0}^{t^{*}}\left(\int_{\tau^{*}}^{t^{*}}\frac{l^{*\prime }\left(\tau^{*}\right)}{\sqrt{t^{*}-\tau^{*}}}d\tau^{*}\right)l^{*\prime }\left(\tau^{*}\right)d\tau^{*},t^{*}<1\\ \int_{0}^{t^{*}-1}\left(\int_{\tau^{*}-1}^{t^{*}}\frac{l^{*\prime }\left(\tau^{*}\right)}{\sqrt{t^{*}-\tau^{*}}}d\tau^{*}\right)l^{*\prime }\left(\tau^{*}\right)d\tau^{*}+\int_{t^{*}-1}^{t^{*}}\left(\int_{\tau^{*}}^{t^{*}}\frac{l^{*\prime }\left(\tau^{*}\right)}{\sqrt{t^{*}-\tau^{*}}}d\tau^{*}\right)l^{*\prime }\left(\tau^{*}\right)d\tau^{*},\;t^{*}>1 \end{array}\right.$$


(15)ψ=eH/πR*Ψ* is the volume ratio of slit to the circular tube, which corresponds to the volume ratio of intra-fiber pores to inter-fiber pores. This reveals the key physical reason for the limited accuracy of the Washburn equation for paper capillary flow. The model agrees well with the experimental results and provides a new explanation for the flow of porous media with size pores.

Cao et al. proposed a mechanism to describe the movement of erythrocytes on paper [163]. The morphology and formation of blood stains on paper substrates and the factors influencing the typical shape of blood stains were investigated at the macroscopic and cellular levels, reporting that the formation of ring-shaped red blood cell stains on paper is mainly the result of a combination of capillary core suction, filtration and evaporation fluxes, and it is influenced by the fiber structure, *RBC* incubation time, relative humidity, and paper additives. The dynamic transport behavior of erythrocytes in paper was quantified by the equation:
(16)$$\rho =\frac{VC_{RBC}}{V_{c}}/\left(\pi \frac{d^{2}}{4}\right)$$
where ρ is the density of red blood cells, d is the wetting diameter, *V* is the volume of *RBC* samples deposited on the paper, *Vc* is the average red blood cell volume of individual red blood cells, and *C_RBC_* is the red blood cell volume concentration.

## 4. Microfluidic Devices Based on Porous Media for Biomedical Analysis

With the aging of the population and the improvement of people’s quality of life, the demand for health testing and medical services has grown rapidly in recent years. In order to effectively solve this problem, the prevention, diagnosis, treatment and management of various diseases have become effective solutions. Biosystems analysis and biomolecular assays in biomedical analysis have become the most important part of healthcare testing management. Traditional medical diagnosis requires expensive specialized testing equipment and complex operations, which can be difficult to access for resource-limited areas. Microfluidic devices, due to their small size, portability, and rapid diagnostic capabilities, offer the potential to significantly reduce costs and expand the scope of medical diagnosis. For example, microfluidic based point-of-care (POC) equipment is used for rapid analysis and testing outside the laboratory [164,165,166]. Wearable and miniaturized real-time detection devices have become a trend in the development of healthcare devices. Microfluidic devices based on porous media offer great potential for next-generation health monitoring and medical services because of their high specific surface area, portability, and affordability.

### 4.1. Porous Media-Based Microfluidic Devices for Biomedical Analysis

Body fluids include tears, sweat, urine, saliva and blood, which are ideal sources for medical diagnostic sampling because of the large number of small biological molecules they contain [7,167]. For example, saliva contains proteins, DNA and many microorganisms, sweat contains many metabolites such as sodium ions, calcium ions, lactic acid and urea, and glucose is present in many body fluids such as blood, sweat and tissue fluids [8].

Saliva, as the most readily available body fluid that does not require in vivo extraction, has become the hottest research area in non-invasive microfluidic devices for diagnosis using body fluid samples [168,169,170]. Saliva is a complex mixture containing various proteins from blood (C-reactive protein, α-1B glycoprotein, etc.) and glucose from sweat, among other components. Therefore, through different biomarkers in saliva, it is possible to reflect the body’s systemic health status (cardiovascular disease, cancer, diabetes, etc.) [171]. Glucose levels in saliva can replace blood as a routine screening tool for diabetes because of its correlation with blood glucose [171,172,173,174]. Therefore, in recent years, microfluidic devices based on glucose detection in porous media have become a research hotspot [175,176,177,178]. Jia et al. first proposed a point-of-care glucose sensor with fiber paper integrated with graphene oxide [179]. The porous structure in the medium effectively increased the absorption rate of the reagent, thus improving the reaction efficiency and the uniformity of the color distribution of glucose. Nitrite is a toxic class 2A carcinogen commonly found in spoiled and expired foods, food additives and preservatives. Therefore, the detection and monitoring of nitrite in humans is essential for the protection of human health. Zhang et al. developed colorimetric microfluidic paper-based analysis devices based on electrokinetic stacking that can be used to detect nitrite concentrations in saliva [180]. Saliva was passed on paper-based microchannels coated with Griess reagent, and the microfluidic paper-based analysis devices was shown to have a linear response range of 0.075–1.0 μg/mL with a limit of detection of 73 ng/mL. Devarakonda et al. developed a microfluidic point-of-care device for detecting influenza virus in saliva samples (Figure 5a). The device was coated with superhydrophobic silica nanoparticles on modified porous fiber paper, and then, the electrodes were patterned by a printing technique using single-walled carbon nanotubes and chitosan modifications. This microfluidic device was shown to have good detectability of H1N1 virus [181]. Izabela et al. probed the lithium ion concentration in saliva by a microfluidic wire-based device. The combination of extractant and porous media was used to effectively improve the sensitivity of lithium ion extraction and device [182].

For non-invasive devices, sweat is another important body fluid to detect because of its biomarkers such as electrolytes, glucose, protein, and lactate [34]. Since sweat is only produced when the body is exercising or when the room temperature is high, the breathability and comfort of microfluidic devices become the direction to focus on sweat [183,184,185,186]. Microfluidic sensors with porous media have good breathability and can collect sweat quickly and efficiently through capillaries. The ion concentration in sweat can reflect the balance of water and various substances in the human body [187]. Maryam et al. presented a microfluidic paper-based analysis device to measure chloride ion content in sweat. A scale in the microchannel was used to quantify adsorbed chloride ions and thus detect cystic fibrosis [188]. Creatine levels in sweat can reflect the recovery of the body after exercise. The POC device is capable of analyzing a small sample (2 μL) within one minute without requiring any electronical-based readout. This makes it a suitable option for providing CF diagnosis in developing countries and resource-limited areas. Curto et al. proposed the preparation of a flexible microfluidic platform based on ionic liquid polymer gel. This portable, wearable microfluidic device can provide real-time feedback on sweat composition during human exercise [31]. Wang et al. reported a highly flexible graphene-based paper platform for the detection of lactate in sweat, consisting of Cu submicron buds deposited on free-standing graphite paper and monolayer molybdenum disulfide crystals [180] (Figure 5b). The sensor displays a linear lactate detection range of 0.01–18.4 mM. 

**Figure 5 micromachines-14-00547-f005:**
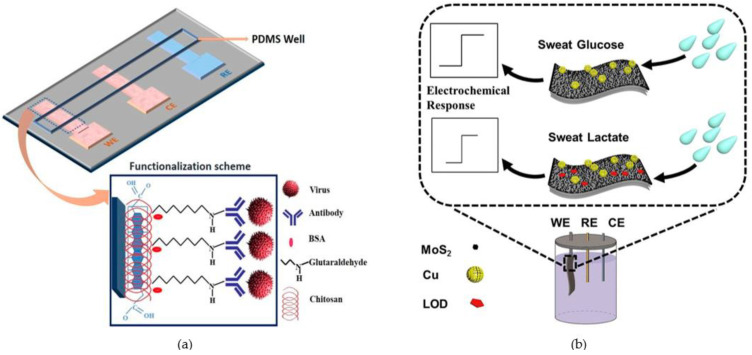
(**a**) The paper-based immunosensor with a PDMS well; (**b**) Graphene paper-based platform for sweat glucose and lactate sensing. (**a**) Reproduced with permission. Ref. [181] Copyright 2017, Sensors (Basel). (**b**) Reproduced with permission. Ref. [189] Copyright 2018, Anal Biochem.

Torul et al. mapped microfluidic channels on the surface of porous cellulose membranes by the wax-printing technique for the preparation of microfluidic devices for the detection of glucose in blood. When blood flows in, a large number of blood cells, etc., will be retained on the nitrocellulose membrane, and small molecules such as glucose enter the detection zone smoothly and are characterized by surface-enhanced Raman scattering spectroscopy [190]. Table 2 briefly reviews several other microfluidic devices proposed in the literature for the detection of porous media in body fluid samples (Table 2).

### 4.2. Microfluidic Devices for Other Biomedical Analysis Applications

Microfluidic devices have important applications in other biomedical aspects in addition to the analysis and detection of body fluid components [7,216,217]. The transport and delivery of specific fluids (drugs, cells, etc.) using microfluidic channels with porous media has become a hot research direction in recent years due to the unique physical and chemical properties of microfluidic channels [218,219,220,221]. Nanoparticles can encapsulate various drugs to improve their stability and solubility, transport and release them through microfluidic channels to make the transport process controllable and reduce their toxicity [222]. For example, Chen et al. developed a PDMS microfluidic chip for generating polymer nanoparticles loaded with curcumin by generating a gas segmented liquid plug. The controllable nanoparticle diameter can be achieved by regulating the flow rate of liquid in the channel [223]. Microfluidic devices also have the ability to capture, align and manipulate cells. Cell separation and capture can be achieved by preparing arrays of porous microstructures, such as separating cancer cells from healthy cells for cancer diagnosis [224,225,226,227]. Yin et al. prepared the first pyramidal porous array to capture specific cells [225] (Figure 6a). In this platform, micro-cavity filter arrays were designed to capture and enrich circulating tumor cells from primary blood samples and ensure the escape of red blood cells and leukocytes. It was demonstrated that less than 0.003% of leukocytes remained in the pyramidal microarrays while ensuring the capture of more than 83% of circulating tumor cells. Microfluidic devices based on porous structures have unique advantages in chemical and biomolecular screening due to their inherent structural properties. Yao et al. used PDMS sub-nanoliter pore arrays to isolate and analyze circulating tumor cells in whole blood, and only a few circulating tumor cells possessed the phenotype of metastatic potential and secreted protein hydrolases [226]. The screening and detection of cells by microfluidics based on porous structures can improve our understanding and analysis of cells; thus, disease screening and development provide more new options. Heavy metals are a group of substances that seriously pollute the environment and cause harm to the human body. For example, lead ions can cause great damage to the kidneys and lead to abnormal neurodevelopment in humans, and cadmium ions can affect the proliferation and differentiation of human cells [228]. Therefore, the detection of heavy metal ions has become an important research direction to protect human health. Rattanarat et al. developed an inexpensive and mass-produced paper-based microfluidic assay device that can detect six different metals by the separation of colorimetric and electrochemical methods [40] (Figure 6b). In the colorimetric detection mode, the detection limits of the platform were 0.12, 0.75, 0.75 and 0.75 μg for Cr, Fe, Cu and Ni, and 0.25 ng (1 μg/L) was obtained for Pb and Cd when analyzed on 2 mm and 10 mm filter punches, respectively. These low-cost portable detection devices can easily and quickly detect toxic metals. Porous materials possess unique properties for fluid flow and separation, and they can also serve as scaffolds to mimic the process of cell proliferation and differentiation in biological tissues for 3D cell culture. This makes them highly versatile for use in various applications, including cell physiology, tumor models, and drug delivery [229,230,231]. Yu et al. developed a PMMA master mold with 3D undulated microtopographies, which was then used to create a PDMS production mold [232]. Gelatin chondroitin sulfate-6 sulfate (Gel-C6S-HA) was then filled into the PDMS mold and freeze-dried to obtain a porous scaffold. Newborn human fibroblasts (NHF) were cultured on the scaffold surface for up to 7 days, demonstrating the biocompatibility of the scaffold and showing unique cell responses at a macroscopic level with biomimetic morphology. Li et al. prepared a method for constructing aligned porous scaffolds for 3D cell culture that does not require cross-linking agents [233]. The porosity of the porous foam was adjusted by controlling the amount of NaCl and the ratio of the oil phase. The foam scaffold does not possess cytotoxicity. Mouse fibroblast cells NIH/3T3 were grown in the surface and internal structures of the foam, which proved its promising application in bioadaptive 3D scaffolds for tissue engineering. Extracellular vesicles derived from tumor cells, which can be stably detected in various body fluids and reflect the tumor burden status in real time, are considered a promising tool for liquid biopsy. Li et al. functionalized a 3D porous PDMS sponge structure with CD9 antibody to capture extracellular vesicles imported into a microfluidic chip [234]. This work is based on a 3D porous microfluidic chip platform and is used as a new non-invasive diagnostic tool for the early detection of colorectal cancer.

## 5. Conclusions and Perspective

This paper reviews the recent advances in microfluidic devices based on porous discontinuous media, covering from the preparation and principles of porous media to the application of the devices in biomedical fields. Microfluidics based on discontinuous media is currently of great value in the biomedical field due to comprehensive and precise theoretical studies and technological innovations. Despite the many exciting research and innovations, many challenges remain.

The first problem should focus on the materials and preparation methods of discontinuous porous media. The existing preparation methods generally focus on technical methods such as photolithography, etching and printing. However, these methods all have relatively more steps and longer preparation cycles, which are not conducive to reducing the cost of the device and the mass production of the device. In addition, most of the current material choices for porous media are focused on PDMS, paper and some etched metals, which cannot have both flexibility and stability in some special environments. Therefore, more new materials with excellent performance need to be explored and applied.

The second issue focuses on the fundamental study of microfluidics. Due to the relatively complex composition of fluids in practical applications, many fluid mechanics with conditional limitations and fundamentals are no longer applicable. The study of surface tension, rheological behavior, interfacial stability and hysteresis effects of impure components needs to be further explored. In addition, the theory of more complex interfacial parameters with time and space needs to be explored more deeply, considering the possible adsorption of certain interfacially active components.

The last issue concerns the practical applications in the medical and healthcare field. Currently, most microfluidic devices based on porous structures are prepared in research institutions or in university laboratories. This firstly limits the access to microfluidics, and the separation between laboratory and commerce prevents most of the research from realizing its market applications and commercial value. Secondly, the laboratory has sophisticated equipment and stable testing environment, which makes many excellent performance and stable data uncontrollable in practical applications. Therefore, there is a need to reconcile research results with commercial manufacturing.

Despite some of the current issues that need to be addressed, microfluidics has found an exciting and widespread use in biomedical analysis. The emergence of advanced technologies such as artificial intelligence, deep learning, and simulation techniques provides tremendous opportunities for the development of microfluidics based on porous media. Artificial intelligence can process and analyze data extremely quickly to assist clinicians in making accurate and timely diagnoses. Deep learning and big data can effectively analyze global user-generated data and make trend predictions. Multi-disciplinary collaboration is also a welcome trend. Numerous research institutions are already producing exciting results through the assistance and deep collaboration of researchers from different fields, which bridges the gap between material synthesis, theoretical research and applications. We firmly believe that the future will bring more exciting achievements in microfluidics based on porous materials.

## Figures and Tables

**Figure 1 micromachines-14-00547-f001:**
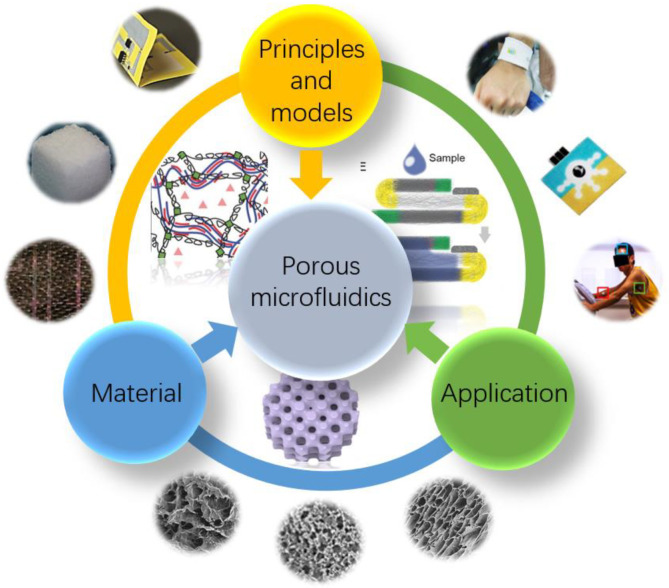
Illustration of the structure of this review. With the help of advanced material and theoretical modeling, the constantly advancing applications of porous microfluidic device for biomedical diagnosis. Reproduced with permission. Ref. [31] Copyright 2012, Sensors and Actuators B: Chemical. Reproduced with permission. Ref. [32] Copyright 2017, Advanced Functional Materials. Reproduced with permission. Ref. [33] Copyright 2017, Angewandte Chemie International Edition. Reproduced with permission. Ref. [34] Copyright 2019, Science Advances. Reproduced with permission. Ref. [35] Copyright 2011, ACS Appl Mater Interfaces. Reproduced with permission. Ref. [36] Copyright 2019, Acta Biomater. Reproduced with permission. Ref. [37] Copyright 2016, Adv Mater. Reproduced with permission. Ref. [38] Copyright 2016, Adv Mater. Reproduced with permission. Ref. [39] Copyright 2012, ACS Appl Mater Interfaces. Reproduced with permission. Copyright 2014, Anal Chem [40].

**Figure 2 micromachines-14-00547-f002:**
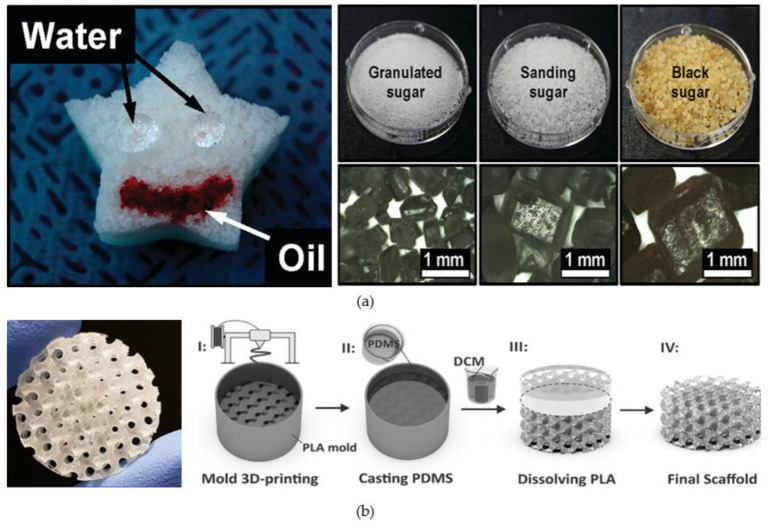
(**a**) Porous PDMS sponges and optical microscope images of various sugar particles; (**b**) Porous PDMS scaffolds and its fabrication process. (**a**) Reproduced with permission. Ref. [35] Copyright 2011, ACS Appl Mater Interfaces. (**b**) Reproduced with permission. Ref. [36] Copyright 2019, Acta Biomater.

**Figure 3 micromachines-14-00547-f003:**
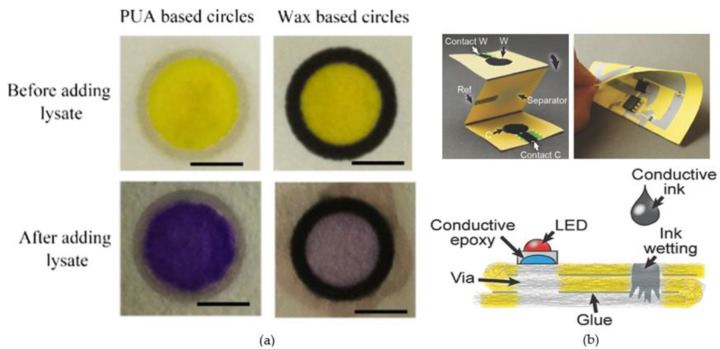
(**a**) Microfluidic paper-based analytical devices for colorimetric assays for *E. coli* BL21; (**b**) Photos and schematic diagram of the paper-based printed 3D circuit integrating electronics and microfluidics. (**a**) Reproduced with permission. Ref. [101] Copyright 2020, Sensors and Actuators B: Chemical. (**b**) Re-produced with permission. Ref. [37] Copyright 2016, Adv Mater.

**Figure 4 micromachines-14-00547-f004:**
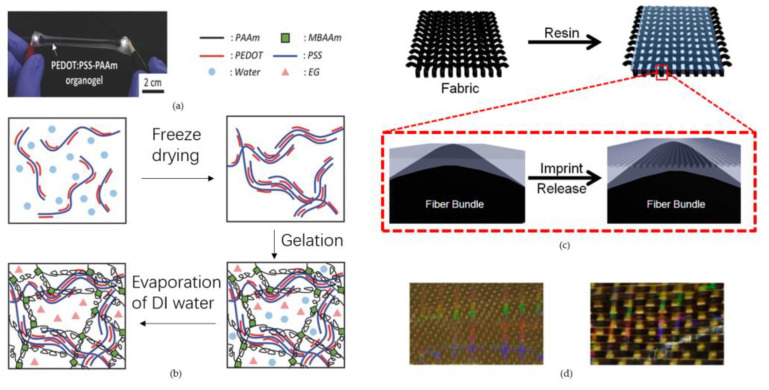
(**a**) Photo of PETDOT:PSS-PAAm organogel stretch up to 200% strain; (**b**) Synthesis procedure of the electrically conductive PEDOT:PSS–PAAm organogels; (**c**) Schematic of the fabrication of a patterned fiber composite; (**d**) Macroscopic photographs of E-glass/x-PDMS and Kevlar-carbon fiber/x-PDMS. (**a**,**b**) Reproduced with permission. Ref. [38] Copyright 2016, Adv Mater. (**c**,**d**) Reproduced with permission. Ref. [39] Copyright 2012, ACS Appl Mater Interfaces.

**Figure 6 micromachines-14-00547-f006:**
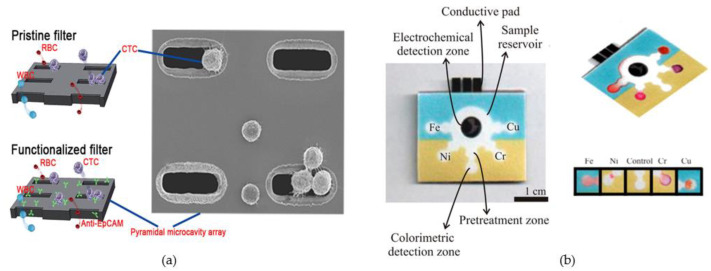
(**a**) Filter process of the pristine filter and the functionalized filter; (**b**) microfluidic paper-based analytical device for toxic metals detection. (**a**) Reproduced with permission. Ref. [225] Copyright 2019, Anal Chim Acta. (**b**) Reproduced with permission. Ref. [40] Copyright 2014, Anal Chem.

**Table 1 micromachines-14-00547-t001:** Comparison of preparation methods and their advantages and disadvantages for different materials.

Material	Advantages	Limitations	Fabrication Technique	Advantages	Limitations	Refs.
PDMS	Easy fabrication, high flexibility, and thermal stability	Poor long-term stability	Soft lithography	High-resolution, facile fabrication	Difficulty in fabrication over large areas	[112,113]
			Sacrificial template	Low-cost, size adjustable	Difficulty in removing templates, uneven pore distribution may exist	[35,36,114]
			Track etching	Allows the formation of uniform pore sizes and controlled pore densities	Time consuming	[115]
			Gas foaming technique	Facile and eco-friendly fabrication procedures	Poor control of pore size and porosity	[116]
			Laser micromachining	Simple, fast and low cast, high precision	The principle of interaction between the material and the laser is not entirely clear	[117,118]
			3D printing	Desirable pore size and porosity	Relatively high cost, low fabrication efficiency	[36,119]
PMMA	Low processing cost, good mechanical properties,	Poor biocompatibility	Hot embossing	Facile, low cast	Requires high temperature and pressure conditions	[86,120]
			Direct laser writing	Short cycle time of production	Limited resolution	[121,122]
			Injection molding	Fast, high production efficiency	High cost of mold equipment	[123]
Paper	Cost-effective, simple, disposable, and portable	Channel size is not easy to control and not standardized, Auto-fluorescence interference	Photolithography	High resolution	Difficulty in fabrication process, time consuming, high cost	[1,124]
			Printing	Low cast, simple operation procedures	Low resolution	[100,125]
			Etching	Low cast	Low resolution and complexity, difficult to fabricate high-density microchannel networks	[126,127]
			Embossing	Complex microfluidic networks can be prepared	The preparation process is complicated	[100,128]
Hydrogels	Biocompatibility, mechanical tenability	Complex preparation process	Templated-assisted	Control of the porous properties, morphology, and structure	Time consuming, complex process of template leaching	[129]
			Freeze drying	Suitable for almost any material	High energy consumption, inability to precisely control porosity	[130,131]
			3D printing	Rapid and can produce complex, three-dimensional structures.	Limited resolution	[132,133]
Textile	Excellent biocompatibility	Non-standardized, not easy to mass produce	Spinning	Facile fabrication	Difficult to precise modeling	[134,135]
			Electrospinning	Can prepare nano-scale microfiber	Inability to precisely control fiber diameter	[20]

**Table 2 micromachines-14-00547-t002:** Summarizes the porous based microfluidics for detection on body fluid samples.

Ref. and Years	Materials	Fabrication Methods	Detection Methods	Target and Sample Matrices	Detection Limit
[170] Patarajarin et al., 2022	Paper	Wax printing	Antigen test	SARS-CoV-2 (Saliva)	1 fg/μL
[184] Hong et al., 2022	Hybrid Janus Membrane	Roller-assisted liquid printing.	Electrochemical	Glucose and lactate (sweat).	0.15 μL
[185] Li et al., 2022	Hydrogel paper	Self-assembled	Electrochemical	Glucose (sweat)	10.3 μM
[186] Mogera et al., 2022	Paper	Cutting	Surface-enhanced Raman spectroscopy (SERS)	Uric acid (sweat)	1 μM
[191] Li et al., 2021	Paper	Printing	Electrochemical	Glucose and lactate (sweat)	17.05 μM
[192] Bagheri et al., 2021	Paper	Wax printing	Electrochemical	Copper ions (sweat and serum)	3 ppb
[193] Fiore et al., 2023	Paper	Waxing printing	Electrochemical	Cortisol (sweat)	101 mM
[194] Weng et al., 2022	Paper	Screen-printing	Electrochemical	Cortisol (sweat)	0.1 nM
[195] Singh et al., 2022	Paper	Cutting	Electrochemical	Glucose (sweat)	0.5 μM
[196] Fabiani et al., 2022	Paper	Wax printing	Electrochemical	SARS-CoV-2 (saliva)	0.1 ug/mL
[197] Moon et al., 2022	PVA-based hydrogel	Sacrificial template	Electrochemical	βHydroxybutyrate (sweat)	62 μM
[198] Gunatilake et al., 2021	Nanotubes alginate hydrogel	Freeze-drying	Colorimetric	Glucose (sweat)	0.8 mM
[199] Guzman et al., 2020	Hydrogel	Sacrificial template	Colorimetric	Lipocalin-1 (tear)	1 ng/mL
[200] Xu et al., 2021	PEDOT:PSS hydrogel	Sacrificial template	Electrochemical	Uric acid (sweat)	1.2 μM
[201] Siripongpreda et al., 2021	Hydrogel	Matrix deposition	Colorimetric	Glucose (sweat)	25 μM
[202] Yeung et al., 2022	Graphene	Chemical vapor deposition	Electrochemical	Na+ (sweat)	10 mM
[203] Yoon et al., 2020	Graphene	Laser-induced	Electrochemical	Glucose (sweat)	300 nM
[204] Wang et al., 2021	Hydrogels	Cross-linking	Strain sensor	NaCl (sweat)	0.15 μL
[205] Saha et al., 2021	Paper	Cutting	Colorimetry	Lactate (sweat)	20 mM
[47] Baretta et al., 2023	Hydrogel	Template	Electrochemical	Glucose (serum)	1 mM
[206] Liu et al., 2021	PDMS	Template	Electrochemical	Cortisol (sweat)	0.3 fg/mL
[207] Li et al., 2023	Graphene	Hydrothermal	Electrochemical	Glucose (sweat)	2.45 μM
[208] Xuan et al., 2018	Graphene	Laser-induced	Electrochemical	Glucose (sweat)	5 μM
[209] Kil et al., 2022	Graphene inks	Printing	Electrochemical	Na+ (sweat)	9.1 × 10^−7^ M
[210] Liu et al., 2021	PEN and SFNFs	Hybridization material strategy	Electrochemical	Glucose (sweat)	2 mM
[211] Xu et al., 2021	Reduced graphene oxide	Electrostatic self-assembly	Electrochemical	Glucose (sweat)	3.7 μM
[212] Poletti et al., 2021	Graphene oxide	Chemical functionalization	Electrochemical	Glucose and lactate (sweat)	32/68 nM
[173] Chakraborty et al., 2020	CuO	Hydrothermal synthesis	Electrochemical	Enzyme-less glucose (saliva)	0.41 μM
[213] Park et al., 2022	Platinum nanozyme-hydrogel composite	Photopolymerization	Colorimetry	Glucose (serum)	3.9 μM
[214] Elancheziyan et al., 2023	Co-PM-NDGPC/SPE	Single-step electrodeposition	Electrochemical	Glucose (blood)	7.9 μM
[215] Yao et al., 2022	ZGC PLNPs	Self-assembly	Fluorescence analysis	Dopamine (serum)	0.001 μM

## Data Availability

Data sharing not applicable No new data were created or analyzed in this study.

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
