# Peer review of "Porous Structural Microfluidic Device for Biomedical Diagnosis: A Review"

_micromachines, 2023, doi:10.3390/mi14030547_

Round 1

Reviewer 1 Report

I would suggest the authors check/revise the whole of the manuscript in terms of proper English writing. E.g. lab on chip should be “lab-on-a-chip”. Line 53: different microfluidics devices should be “various microfluidic devices.  Moreover, authors must know the difference between microfluidics and microfluidic and use the proper word in the text.

I would strongly suggest deleting or moving the 3.1 part. “classical fluid dynamics in the microfluidic platform” after the introduction. Please be aware this manuscript is about porous microfluidic devices for biomedical applications and the description of fluid mechanics is inappropriate and redundant. It is better to remove this part and add some diagnosis models.

Why PMMA has not been described? It is a polymer like PDMS, especially in large-scale fabrication.

The authors must present various devices for biomedical applications such as microfluidic devices for early cancer diagnosis, 3D-cell culture, point-of-care detections…

Reviewer 2 Report

The review paper by Chen et al. provided a comprehensive overview of porous material-based microfluidics, from the aspects of fabrication, physical principle to bioapplications. In general, the topic may attract broad interest and the paper is well-organized. I suggest acceptance after minor revision. Please find below some comments.

1. Part 2 introduced different porous materials as well as their fabrication. It would be nice to add a paragraph (with a table) to compare the properties, advantages and disadvantages of these materials. Additional, I suggest to change the subtitle of 2.1 to the same format as that of 2.2 and 2.3.

2. In Table 1, the detection methods should be referred to as ‘colorimetric’, ‘electrochemical’ instead of ‘smartphone……’‘biosensor’ to provide information on the techniques used for sensing. It would also be nice to add comparisons and discussions on the different techniques based on porous materials, e.g. which technique is better suited for hydrogel-based sensors.

3. Pay attention to the use of capital letters in Table 1. Both ‘glucose’ and ‘Glucose’ were presented.

Round 2

Reviewer 1 Report

The authors performed my requested comment.